# Adiposity Phenotypes and Associated Cardiometabolic Risk Profile in the Inuit Population of Nunavik

**DOI:** 10.3390/nu16050725

**Published:** 2024-03-02

**Authors:** Fannie Lajeunesse-Trempe, Marie-Eve Piché, Paul Poirier, André Tchernof, Pierre Ayotte

**Affiliations:** 1École de Nutrition, Université Laval, Québec, QC G1V 0A6, Canada; fanny.lajeunesse-trempe.1@ulaval.ca (F.L.-T.); andre.tchernof@criucpq.ulaval.ca (A.T.); 2Institut Universitaire de Cardiologie et de Pneumologie de Québec, Québec, QC G1V 4G5, Canadapaul.poirier@criucpq.ulaval.ca (P.P.); 3Faculté de Pharmacie, Université Laval, Québec, QC G1V 0A6, Canada; 4Département de Médecine, Université Laval, Québec, QC G1V 0A6, Canada; 5Département de Médecine Sociale et Préventive, Université Laval, Québec, QC G1V 0A6, Canada; 6Centre de Recherche du CHU de Québec-Université Laval, Québec, QC G1V 0A6, Canada; 7Institut National de Santé Publique du Québec, Québec, QC G1V 5B3, Canada

**Keywords:** obesity, indigenous, Inuit, cardio-metabolic risk factors

## Abstract

The Inuit population of Nunavik is faced with a significant rise in the prevalence of obesity [body mass index (BMI) ≥ 30 kg/m^2^], but the impact on cardiometabolic health is unclear. The aim of this study was to characterize adiposity phenotypes and explore their associations with cardiometabolic risk factors among Nunavimmiut men and women. We used data obtained from 1296 Inuit who participated in the *Qanuilirpitaa*? 2017 Nunavik Inuit Health survey. Collected information included demographics, anthropometric measurements including visceral fat level (VFL) measured using electrical bioimpedance, biomarkers, hemodynamics, medical history and medication list. Adjusted population-weighted linear regressions were conducted to assess associations between body fat distribution and cardiometabolic risk factors. The accuracy and cut-off points of anthropometric indices to detect cardiometabolic abnormalities was evaluated by area under the receiver operator characteristic curve (AUROC) and a maximum Youden index analysis. Among Nunavimmiut (mean age 38.8 years [95%CI: 38.4 to 39.3]), obesity was observed in 42.8% of women and 25.6% of men. Compared to men, women presented a higher prevalence of abdominal obesity (78.8% vs. 46.4% in men, *p* < 0.05) and elevated VFL (54.4% vs. 20.1% with an InBody level ≥ 13, *p* < 0.05). Indices of global fat distribution and abdominal adiposity including VFL provided poor to moderate ability to detect cardiometabolic abnormalities (AUROC between 0.64 and 0.79). This analysis revealed that despite a high prevalence of abdominal obesity, particularly among women, anthropometric measurements of adiposity are inconsistently associated cardio-metabolic risk factors in Inuit adults of Nunavik.

## 1. Introduction

Over recent decades, Canadian Indigenous communities have been faced with a significant and disproportionate rise in the prevalence of obesity [body mass index (BMI) ≥ 30 kg/m^2^] [1,2,3] and cardiovascular (CV) risk factors such as hypertension [4,5] and dyslipidemia [6]. Among other factors, the progressive impacts of colonization have contributed to a change in their traditional lifestyle behaviors and diet. This is associated with adverse effects on their CV health and promotes health disparities among Indigenous peoples across Canada [1]. Despite their younger average age, certain Indigenous peoples living in Canada (First Nations, Métis) present a higher prevalence of CV risk factors (insulin resistance, type 2 diabetes, hypertension, dyslipidemia) and exhibit higher CV morbidity and mortality compared to non-Indigenous Canadians [3]. However, reports over the last few decades tend to distinguish the Inuit population from other Indigenous populations in Canada, and suggest a differential effect of fat accumulation on cardiovascular risk factors, mainly insulin resistance and type 2 diabetes (T2D) [7].

The cardiometabolic risk profile of Inuit adults living in Nunavik has been previously described in two major health surveys (the *Santé Québec* Survey in 1992 and the *Qanuippitaa*? Inuit Health Survey in 2004) which included participants from all 14 communities of Nunavik. Results from those surveys have described an increase in the prevalence of obesity (from 19% in 1992 to 28% in 2004), which was encountered particularly in women (from 25 to 31%) [1]. This trend was also observed in Inuit living with severe obesity (BMI ≥ 35 kg/m^2^) [2]. We have previously shown that the prevalence of severe obesity had increased from 1.5% in 1992 to 5.5% in 2004, with women presenting a significant increase in waist circumference (WC) over time (from 103.9 to 121.5 cm), i.e., abdominal obesity [2].

Numerous studies have reported that abdominal obesity and more specifically abdominal visceral fat level (VFL) has an important prognostic role in obesity-related cardiometabolic complications [8,9]. Across the entire spectrum of BMI values, the presence of cardiometabolic conditions appears to be closely related to a predominantly visceral abdominal body fat deposition pattern, as well as ectopic fat accumulation in organs such as the liver and the heart [9]. Although abdominal obesity is now an established universal marker of cardiometabolic risk, the majority of studies supporting this hypothesis did not include Indigenous populations [10,11]. Moreover, despite a significant rise in the prevalence of obesity and abdominal obesity over recent decades, the prevalence of cardiometabolic risk factors (dyslipidemia, dysglycemia, insulin resistance, hypertension, liver steatosis, etc.) appears to be relatively stable in Inuit men and women, reinforcing the idea that the Inuit population would benefit from a more favorable cardiometabolic risk profile, independent of their degree of obesity [2]. Cross-sectional studies including Inuit of the central Canadian Arctic (434 participants, 54.1% women, mean BMI of 24.7 kg/m^2^ in men and 28.6 kg/m^2^ in women) [7] and Nunavimmiut (710 participants, 54.4% women, BMI of 27.1 kg/m^2^ in men and of 28.0 kg/m^2^ in women) [6] have reported a weaker association between anthropometric indices of abdominal obesity (WC) and cardiometabolic risk factors, compared to non-Indigenous populations. In a study using data from the *Qanuilirpitaa*? 2017 survey (1177 participants, 65.4% women and 32% of obesity), we reported that abdominal obesity (WC ≥ 88 cm in women and ≥102 cm in men) was a risk factor for hypertension, but this association was weaker in women compared to men, suggesting that metabolically healthy obesity might be more prevalent among Inuit women [4]. These results raise important questions about the generalization of pathophysiological concepts associating abdominal obesity and detrimental cardiometabolic health to Inuit populations. This also challenged the applicability of our current obesity public health policies and CV clinical guidelines [12,13,14] to the Inuit population of Nunavik.

This study is the first to explore the association between measured indices of body fat distribution including VFL level and cardiometabolic health using contemporary data in Inuit from Nunavik. The aim of this study was (1) to characterize adiposity phenotypes and the cardiometabolic risk profile among Inuit men and women from Nunavik; (2) to describe the association between anthropometric indices and cardiometabolic risk factors; and (3) to explore sex- and ethnic-specific cut-offs for global obesity and abdominal obesity in the Inuit population of Nunavik.

## 2. Methodology

### 2.1. Setting

The *Qanuilirpitaa*? 2017 Nunavik Inuit Health survey was conducted in partnership with Nunavik health organizations, the *Institut National de Santé Publique du Québec* (INSPQ) and research teams from different universities in Canada. It took place following a resolution adopted by the Nunavik Regional Board of Health and Social Services (NRBHSS), which requested updated information on the overall health of the Nunavik population. An Inuit-led Steering Committee oversaw the preparation, conduct, interpretation and dissemination of all survey results. The Data Management Committee of *Qanuilirpitaa*? 2017 evaluated the relevance of our research question for the Nunavik population, approved data and biological sample requests, made comments on this manuscript and approved its final version. The dissemination of the results presented in the present study was prepared according to the recent call for action on culturally safe research in Canadian Indigenous populations by Aziz et al. [15]. The *Qanuilirpitaa*? 2017 survey was also approved by the ethics committee of the Centre de Recherche du CHU de Québec—Université Laval (no. 2016-2499-21).

### 2.2. Study Population

Data collection for the *Qanuilirpitaa?* 2017 health survey took place in the 14 communities of Nunavik (Northern Quebec, Canada), from 19 August to 5 October 2017. Participants aged 16 years and older were invited aboard the *Amundsen*, a Canadian Coast Guard icebreaker, converted into a scientific research ship. Potential participants were selected from a proportional sampling of the Makivvik Corporation’s beneficiary list of Inuit living in Nunavik, stratified for age groups and communities. The Makivvik list is the most comprehensive available list to use as a sampling frame for research purposes. It is estimated that the majority of Inuit individuals living in Nunavik are on this list, and for this health survey, the most recent list (spring 2017) has been obtained. The overall participation rate was 36.5%; many individuals originally selected from the list could not be contacted as they were out of the community during the study period. The inclusion criterion was set at 16 years of age or older. This specific age threshold was established based on the community request to include youth for the purpose of assessing their overall health. The only pre-established exclusion criterion for this study was pregnancy (n = 30). A total of 1326 individuals were eligible and included in the *Qanuilirpitaa?* 2017 health survey, and 1296 were included in this specific study. Aboard the *Amundsen*, medical and lifestyle questionnaires were administered and clinical and laboratory tests were performed. Some information including medication lists (n = 1047) was obtained from participants’ medical records in local community-service centers with the participants’ authorization. The detailed methodology has been described elsewhere [16].

### 2.3. Clinical Assessment, Medical Review and Questionnaires

Weight and body composition measurements based on bioelectrical impedance were obtained using a validated InBody device (Model 570, Cerritos, CA, USA) [17]. BMI was calculated by dividing the weight in kilograms by the square of the height in meters (kg/m^2^), and classified according to the National Institute of Health’s scheme (BMI value 25.0–29.9 kg/m^2^ being overweight, 30.0–34.9 kg/m^2^ being obesity, and ≥35.0 kg/m^2^ being severe obesity) [18]. Visceral fat level (VFL) was provided by the InBody device and a high level of visceral fat was defined as per the manufacturer’s recommendations and previous validation studies (level ≥ 13) [17,19,20], which did not include Canadian Inuit populations. To estimate visceral fat level using the InBody device, participants had to stand on the monitor without shoes. A low and safe electrical signal is sent from the feet to the abdomen to produce body composition measurements. Height and WC were measured in centimeters and taken by nurses or trained interviewers and the waist-to-height ratio (WHtR) was calculated by dividing the participant’s WC by its height (in cm). Due to the lack of published data regarding Inuit people, abdominal obesity was defined using the current World Health Organization (WHO) recommended cut-offs of WC (≥88 cm in women and ≥102 cm in men) [21]; however, it is possibly not appropriate for non-Caucasian ethnic groups [22,23]. Resting blood pressure was measured with the ProBP 2400 Digital electronic device (Welch Allyn, Aurburn, NY, USA) according to the 2005 Canadian Hypertension Education Program recommendations. Participants first rested for 5 min in a seated position and the blood pressure cuff was placed on the supported bare arm at the heart level. Three measurements were taken for the same arm with the participant in the same position, but the first reading was discarded, and the blood pressure recorded was the mean of the two latter measures. Hypertension was defined as a systolic blood pressure ≥ 140 mm Hg or a diastolic blood pressure ≥ 90 mm Hg or if participants were on anti-hypertensive medications [24].

Blood samples were collected by trained nurses and processed by laboratory technicians within 90 min of collection. White blood cell count (WBC), platelet count (PLT) and hemoglobin (Hb) concentration were obtained using a DxH 500 hematology analyzer (Beckman Coulter). In addition, the following clinical biochemistry tests were conducted in collaborating hospital laboratories on samples stored at −80 °C: hepatic profile [albumin, aspartate transaminase (AST), alanine transaminase (ALT), gamma-glutamyl-transferase (GGT) and total bilirubin], renal function (creatinine, eGFR, urine micro-albuminuria), lipid profile [triglycerides (TG), low-density-lipoprotein cholesterol (LDL-C), total cholesterol (TC), high-density-lipoprotein cholesterol (HDL-C) and apolipoprotein-B (ApoB)], glycated hemoglobin (HbA1c) and high-sensitivity C-reactive protein (hs-CRP). When participants were fasting (n = 548), fasting glucose and fasting insulin were also measured. The Homeostatic Model Assessment was used to obtain a quantitative assessment of insulin resistance (HOMA-IR) [25]. Insulin resistance was defined as a HOMA-IR index > 2.0 according to previous results reported in a First Nation population of Northern Quebec (Eeyou Istchee Crees) [26].

T2D and pre-diabetes were defined using criteria from Diabetes Canada [27] or determined according to the participant’s medication (i.e., participants on glucose-lowering drugs were considered to be diagnosed with T2D). Dyslipidemia was diagnosed using the Canadian Cardiovascular Society’s recommendations or if participants were on lipid-lowering drugs [28] and stages of chronic kidney disease were reported as per the Kidney Disease: Improving Global Outcomes recommendations [29]. To estimate non-alcoholic fatty liver disease (NAFLD), the fatty liver index (FLI) was calculated using the formula elaborated by Bedogni and colleagues with recommended cut-offs [30], although this index has not been previously validated for Inuit populations. The FIB-4 score was also used to estimate the level of hepatic fibrosis and advanced fibrosis; scores > 2.67 have demonstrated a high positive predictive value for advanced liver fibrosis and this cut-off has been used previously in the Greenlandic Inuit population [31,32,33]. Standardized and culturally appropriate questionnaires were administered to participants to collect information on their lifestyle (smoking status and alcohol consumption).

### 2.4. Statistical Analysis

All statistical analyses were conducted using StataCorp 15.0. Normally distributed continuous variables were reported using mean and standard deviation (SD) and compared between sexes using Student’s *t*-tests. Categorical variables were reported as frequencies and compared with Chi-square tests. Associations between anthropometric measurements of fat distribution (weight, BMI, WC and WHtR) and measurements of fat distribution using electrical bio-impedance (estimated percentage of body fat, body fat mass and VFL) as well as cardio-metabolic risk factors were assessed using logistic and linear regression analyses, adjusted for age, gender, smoking status and BMI in the case of predictor variables WC and VFL, given the previously estimated collinearity. The data employed in the logistic and linear regression analyses underwent prior assessments to ensure normality and linearity, with all assumptions being satisfied. As no cut-off for indices of adiposity have been validated in the Inuit population, independent variables of body phenotypes (weight, BMI, WC and WHtR) were analyzed as continuous variables. Capacities of weight, BMI, WC, WHtR and VFL to predict cardiometabolic diseases were calculated using the area under the receiver operating characteristic curves (AUROCs). An AUC of 0.500 denoted no discriminatory power, 0.501–0.69 denoted poor discriminatory ability, 0.700–0.79 was considered acceptable discriminatory ability, 0.800–0.89 denoted excellent discriminatory ability, and 0.90–1 was considered outstanding discriminatory ability [34]. A Youden index analysis was used to determine the optimal cut-off for the AUROCs [35]. Two-sided *p* < 0.05 was considered statistically significant. Survey weights were applied in all analyses to consider the sampling methodology and nonresponse [16], and bootstrap weights (500) were used to compute the variance of all estimates.

## 3. Results

Baseline characteristics of participants are described in Table 1. A total of 843 (65.0%) women were included in this study, and the majority of participants were under 50 years old (79.9% of men and 82.7% of women), with a large proportion under 19 years old (30.0% of men and 26.6% of women). Mean BMI was 26.3 ± 5.7 kg/m^2^ in men and 28.0 ± 6.4 kg/m^2^ in women (*p* < 0.05). Obesity was noted in 116 men (25.6%) and 361 women (42.8%), among which 22 men (4.9%) and 144 women (17.1%) presented with severe obesity. Severe obesity was more prevalent among young Nunavimmiut women (between 16 to 44 years old) (Figure 1). Inuit women presented a higher prevalence of abdominal obesity (as defined by WHO cut-offs) compared to men (78.8 vs. 46.4%, *p* < 0.05). Furthermore, Nunavimmiut women also had higher levels of VFL (estimated using electrical bio-impedance) compared to men (54.4 vs. 20.1%, *p* < 0.05).

Prevalence of obesity-associated comorbidities are also displayed in Table 1. Nunavimmiut women presented a lower prevalence of hypertension and dyslipidemia compared to men (15.2 vs. 25.8% for hypertension and 29.5 vs. 36.2% for dyslipidemia, *p* < 0.05, respectively). Insulin resistance (estimated using HOMA-IR) was highly prevalent among men and women (55.4% of men and 67.3% of women, *p* = 0.09), but prevalence of prediabetes and T2D (diagnosed using fasting glucose and/or HbA1c) was relatively low for both sexes (9.1% of men and 10.0% of women had prediabetes, 6.0% of men and 7.3% of women had T2D, *p* < 0.05). Using the FLI score, only a small proportion of men and women were diagnosed with NAFLD (3.1 and 2.4%, *p* < 0.05, respectively) and only two participants [one man (0.22%) and one woman (0.12%), *p* < 0.05] were characterized with advanced hepatic fibrosis, defined with the FIB-4 score. Tobacco consumption was high in both sexes, with 64.5% of men and 70.2% of women reporting daily smoking.

Clinical biochemistry data are presented in Table 2. Overall, laboratory features of cardiometabolic risk factors were similar between sex, except for HbA1c, which was significantly higher in men compared to women (HbA1c of 5.56 ± 0.60% in men vs. 5.47 ± 0.74% in women, *p* < 0.05).

Correlations between anthropometric parameters (BMI, WC, WHtR and VFL) were estimated using standardized linear regression analysis (adjusted for age and BMI in the case of WC and WHtR). Correlation factors with VFL were r = 0.90 for BMI, r = 0.87 for WC and r = 0.89 for WHtR (*p*-value <0.05 for all). Estimated predicted values of VFL using anthropometric measurement of adiposity are displayed in Figure 2. Anthropometric indices were all significantly correlated with VFL, with BMI being more strongly associated with VFL (standardized β-coefficient of 5.19, 95% CI: 5.04–5.29), compared to WC (standardized β-coefficient of 1.11, 95% CI: 0.84–1.39) and WHtR (standardized β-coefficient of 0.60, 95% CI: 0.31–0.90). For similar values of BMI and WC, women consistently presented a greater adjusted predicted VFL value compared to men. This trend was also noted for WHtR; the estimated VFL value was higher in women, but only up to a certain WHtR level (WHtR = 1.3).

Associations between the anthropometric features of obesity (weight, BMI, WC, WHtR and VFL) and cardiometabolic risk factors (TC/HDL ratio, LDL, HDL, TG, ApoB, HbA1c, fasting glucose and fasting insulin, HOMA-IR, systolic and diastolic blood pressure) are displayed in Appendix A, with subgroup analyses for men (Appendix A) and for premenopausal women (Appendix A). In men, BMI was associated with a detrimental increase in almost all cardiometabolic risk factors (decrease in case of HDL), but HbA1c. Men demonstrated no significant association between measurements of abdominal obesity and a detrimental cardiometabolic risk profile. Although VFL was associated with an abnormal lipid profile (higher TG and CHOL/HDL ratio) and a higher level of fasting glucose, it was neither significantly linked to an increase in LDL, ApoB, a decrease in HDL nor to higherHOMA-IR, HbA1c and systolic blood pressure values. Specifically looking at premenopausal women, WC and WHtR were negatively associated with insulin resistance and positively associated with dyslipidemia (high TG and ApoB levels). VFL was associated with an abnormal lipid profile and higher diastolic blood pressure but not with risk factors of insulin resistance or dysglycemia. Therefore, in premenopausal Nunavimmiut women, VFL was not associated with any cardiometabolic risk factors albeit accumulating more VFL than Nunavimmiut men. A subgroup analysis excluding participants under the age of 20 showed similar associations compared to the overall included population (Appendix A).

The predictive abilities of anthropometric measurements (weight, BMI, WC, WHtR) and VFL) to detect obesity-associated comorbidities were estimated using AUROC (Table 3). In men, all measurements of fat distribution demonstrated an acceptable capacity to predict hypertension, insulin resistance, prediabetes, T2D and dyslipidemia. However, in women, anthropometric measurements were generally poor predictors of cardiometabolic conditions, except for insulin resistance for which anthropometric indices of obesity showed an acceptable capacity.

The Youden Index analysis was used to determine the optimal sex-specific cut-offs for BMI and WC among Nunavimmiut (Appendix A). In men, optimal BMI cut-offs for cardiometabolic disorders ranged between 25.6 and 28.8 kg/m^2^, while in women, BMI cut-offs were estimated between 26.6 kg/m^2^ and 29.9 kg/m^2^. WC cut-offs for men ranged from 86.5 cm for dyslipidemia and hypertension to 100 cm for prediabetes and diabetes, while WC cut-offs for women ranged from 92.0 cm to 100 cm depending on the outcome. Cardiometabolic comorbidities were optimally detected at lower levels of VFL in men (from 6 to 10) compared to women (from 12 to 14).

## 4. Discussion

In this study, we report a very high prevalence of obesity, severe obesity and abdominal obesity, especially among Nunavimmiut women. Previous surveys from Canada [36] and other countries [37,38] have also described a higher prevalence of abdominal obesity in diverse ethnic female groups compared to males. Comparing our results to a recent study including data from a subgroup of 4662 participants of the Canadian Health Measures Survey Cycles 1 and 2 (mean age of 55.1 years for men and 55.8 years for women) [39], we also demonstrate that Inuit women, despite being younger, have a higher WHtR compared to their Canadian counterparts (0.61 ± 0.10 vs. 0.56 ± 0.10), whereas Inuit men had lower values (0.55 ± 0.10 vs. 0.57 ± 0.08). Our analysis also revealed that, despite being younger (median age [IQR] of 38 [25–63] years old), 12.8% of men and women were characterized by severe obesity, an increased proportion (more than double) compared to the prevalence reported in a weighted analysis from 2004 (5.5% of the total population, median age [IQR] of 48 [28–63] years old) and from 1992 (1.5% of participants, median age of 47 [28–61] years old) [2]. Our results also suggest that obesity prevalence is high in Nunavimmiut youth (16–44 years old). Obesity prevalence in the 16 to 44 years age group varies from 4.5 to 38.2% in men and from 11.8 to 38.8% in women, which is more than twice higher compared to their non-Indigenous counterparts. In comparison with young non-Indigenous Canadians of a similar age (<35 years old), between 7.2 and 15.9% of men and 5.5% to 13.3% of women were characterized with obesity in 2018 [40]. Abdominal obesity is found to be highly prevalent among Inuit men and even more so in Inuit women (46 and 79%, respectively), especially among youth. This trend is particularly worrisome from a long-term perspective, given the risk of morbidity and mortality associated with high-risk obesity observed in non-Inuit populations.

Differences in the association between obesity and cardiometabolic risk factors based on ethnic origin are increasingly noted in the scientific literature and could be partly explained by an ethnic-specific predisposition to store fat subcutaneously compared to viscerally [41,42,43]. A previous study comparing body fat distribution among different ethnic groups suggested that Greenlandic Inuit men and women might accumulate less VAT (measured using ultrasound) for a given level of BMI compared to European-descent populations [42]. However, in our study, we noted high levels of estimated VFL among women and observed a high correlation between anthropometric measurements of body fat (BMI, WC and WHtR) and VFL in both sexes. For a given BMI, WC or WHtR, women almost systematically present higher age-adjusted predicted values of VFL compared to Inuit men (Figure 2). Despite methodological limitations imposed by the use of bio-impedance VFL measurements, this trend is consistent with a previous study, which compared anthropometric measurements between the Greenlandic Inuit (3083 participants, 60.0% women and mean age of 42 years) and a European-descent Danish population (795, 56.4% woman participants and mean age of 47 years), which showed that Inuit women had the highest absolute values in BMI, WC, and VAT accumulation (estimated using standardized measures of ultrasonography) compared to Inuit men and individuals of European descents [41]. Menopausal status was not assessed in the *Qanuilirpitaa*? 2017 health survey but extending our analysis to a subgroup of women aged ≤ 55 years old, we report a similar proportion of women with a high level of VFL (47.1 vs. 54.4% of total women) and age-adjusted predictive values of VFL. If menopause plays a role, its impact on fat distribution among Inuit women remains unclear and will need to be assessed specifically.

Comparing our results to those of the Canadian Health Measures Survey Cycles 1 and 2 (2012–2013), we report that despite a high prevalence of obesity and abdominal obesity, Nunavimmiut have a slightly lower prevalence of dyslipidemia (36.2% in men and 29.5% in women vs. 38% in the Canadian population) [44] and T2D (5.9% in men, 7.3% in women vs. 7.9%) [45] and a higher prevalence of hypertension in men but lower in women (25.3% in men and 15.2% in women vs. 22.6%) compared to the general Canadian population [46]. Our results also indicate a significantly lower prevalence of estimated NAFLD (3.1% in men and 2.4% in women using FLI formula) compared to that reported in the Canadian population (20.8% of Canadians were projected to have a diagnosis of NAFLD in 2019 using a Markov model using historical trends in obesity prevalence and diagnosis of liver fibrosis, decompensated cirrhosis, hepatocellular carcinoma and liver transplantation) [47]. No Canadian data on NAFLD prevalence have been published yet.

It should be noted that the population included in *Qanuilirpitaa*? 2017 is younger than the general Canadian population. Rather than demonstrating a favorable ethnic-specific metabolic profile, our results might illustrate that Inuit men and women from Nunavik are currently characterized by a transient phenotype of obesity, with a short duration of fat accumulation and a preserved metabolic state. After performing analysis on a sub-group of adults 50 years of age and older (Appendix A), we report a prevalence of 28.8% of women with obesity and 75.1% with abdominal obesity, compared to 31.6% of men with obesity and 64.0% with elevated WC. Our analysis also demonstrates that some adiposity-associated comorbidities are more prevalent in this age group. Insulin resistance was diagnosed in the majority of participants (55.4% of men and 67.3% of women, *p* = 0.9). Longitudinal studies will be necessary to better understand how obesity will impact the cardiometabolic health of this group over the long term.

Recent imaging studies have shown that ethnicity could affect metabolic responses to obesity [48,49], and demonstrated that African Americans were less prone to developing a dysglycemic status compared to persons identifying as Hispanic or of European descent, despite a similar level of VAT [49]. Inuit have never been included in large standardized international imaging studies, but specific ethnic disparities could partly explain their low prevalence in cardiometabolic comorbidities, despite a high level of abdominal obesity. When extending our analysis to the effect of abdominal fat accumulation on the cardiometabolic profile of Inuit men and women, we observe a positive association between estimated VFL and risk factors for dyslipidemia and hypertension. However, VFL is not significantly associated with features of dysglycemia (for both sexes combined and in men and premenopausal women separately) and is found to be a poor predictor of pre-diabetes and T2D in Nunavimmiut women (AUROC of 0.69 for both). Similarly, WC was positively associated with certain cardiometabolic parameters (TG, CHOL/HDL, ApoB) but not others (LDL-C, HDL-C, HbA1c, fasting glucose, HOMA-IR, blood pressure) (Appendix A). Beyond body fat distribution, other factors seem to play a significant role in the complex physiological interactions connecting obesity and glucose homeostasis in the Inuit of Nunavik. Among other factors, the traditional diet of Nunavimmiut (low in glucose and rich in protein and long-chain n-3 polyunsaturated fatty acids) might afford protection against cardiometabolic disorders including T2D [50,51,52]. Significant differences in Inuit’s microbiota [53], physical activity level [11], exposure to persistent organic pollutants [52,54] and genetic variants [51] might also impact significantly on their cardiometabolic health. For example, a population-specific nonsense genetic variant in TBC1D4, which has a large impact on T2D risk, has recently been identified in Greenlandic, Canadian and Alaskan Inuit [51,55]. Mutations on TBC1D4 are associated with higher post-prandial glucose and insulin levels. In a recent study, Manousaki et al. have reported that the TBC1D4 mutation was present in 27% of Canadian and Alaskan Inuit (heterozygote or homozygote carriers) [55], which suggests that pre-diabetes and T2D might be underdiagnosed among Canadian Inuit, unless post-prandial values are tested. The prevalence of TBC1D4 mutations have not been yet reported among Nunavimmiut, and further studies examining the validity and diagnostic capacities of traditional screening methods to detect dysglycemia and T2D risk factors are needed in this population.

Although being the most recognized measurement to estimate the degree of obesity and predict associated cardiometabolic disorders, it has been previously suggested that BMI would be inadequate for the Canadian Inuit population [56,57]. Charbonneau-Roberts et al. have previously reported that Inuit men and women tend to have shorter legs and a relatively higher sitting height than other populations, which could lead to an over-estimation of obesity in this population [57]. In a cross-sectional study including 3083 Inuit from Greenland and 795 Europeans from Denmark, Ronn et al. reported a lower percentage of body fat in Greenlandic men and women, despite having higher anthropometric indices of obesity than Europeans [42]. In our study, we observed significant positive associations between BMI, WC, WHtR and estimates of body fat composition and fat distribution using electrical bio-impedance (percentage of body fat, body fat mass and VFL) (Appendix A). However, we also noted poor to moderate capacities of anthropometric indices of obesity to predict cardiometabolic abnormalities (AUROC between 0.60 and 0.77). This suggests that BMI, WC, WHtR and even VFL estimated using electrical bio-impedance are poor predictors of cardiovascular disorders in this population. However, the young age of our population, thus at lower risk for metabolic and cardiovascular disease, needs to be highlighted here and could partly explain these results.

### Strengths and Limitations

This study is the first to extensively describe the cardiometabolic risk profile of Nunavimmiut. Because Nunavik is a remote region of northern Quebec, which is not easily accessible, inclusion of Nunavimmiut in clinical cardiometabolic trials has been limited. This study tentatively addresses the current gap in the scientific literature and provides new data on the metabolic and cardiovascular consequences of body fat accumulation in this population. However, our results need to be interpreted with caution due to certain limitations. First, although all models were adjusted for participants’ age, Nunavimmiut participants in this study were relatively young, which might have impacted on: (1) the prevalence of the cardiometabolic comorbidities observed, and (2) the association between body fat accumulation and cardiovascular and metabolic risk factors. Causal inference of the relationship between adiposity and cardiometabolic health should be examined further in longitudinal studies. Second, measurement of VFL using electrical bio-impedance is not considered the gold-standard and has never been validated in the Inuit population of Nunavik. To our knowledge, no previous studies have reported on VFL using an InBody device in this population, and validation studies will be necessary. Likewise, the cut-off values employed in this study for assessing abdominal obesity lack validation within the Inuit population. It is imperative to undertake more extensive in-community projects, specifically involving under-represented populations like Inuit, to validate commonly used indices and clinical tools for assessing cardiometabolic health. Finally, we employed the Makivvik Corporation’s beneficiary list as a sampling framework to randomly select Inuit participants residing in Nunavik. While it is believed that a significant portion of the population is registered on this list, we acknowledge the possibility that a selection bias may have been induced by the recruitment process.

## 5. Conclusions

In conclusion, Nunavimmiut have a high prevalence of potentially high-risk obesity (abdominal visceral obesity and severe obesity), especially among young women. Early detection and management of cardiometabolic risk factors remain paramount for improving CV health and potentially reducing premature CV mortality. Long-term effects of obesity at a younger age on CV morbidity and mortality risk later in life need to be explored with longitudinal cardiac and whole-body imaging technique studies. Public health actions are required to target this emergent epidemic, but further research is needed to better understand the complex interplays between body fat accumulation and cardiometabolic disorders, in order to develop specific and culturally-appropriate interventions for Nunavimmiut.

## Figures and Tables

**Figure 1 nutrients-16-00725-f001:**
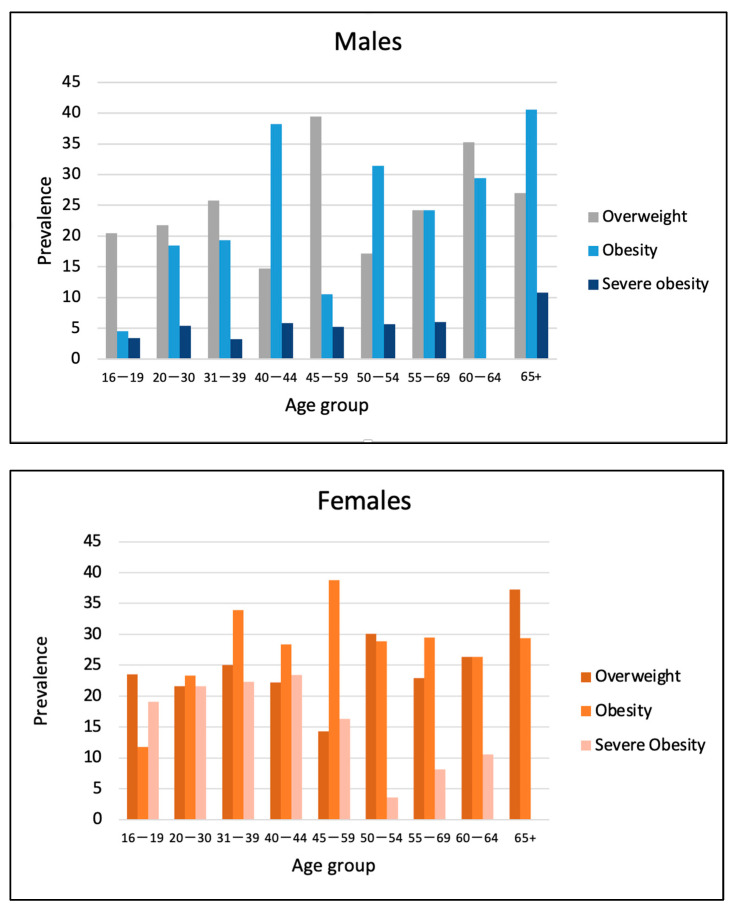
Prevalence of overweight, obesity and severe obesity by sex among Nunavimmiut aged 16 and over, *Qanuilirpitaa*? 2017 Health Survey.

**Figure 2 nutrients-16-00725-f002:**
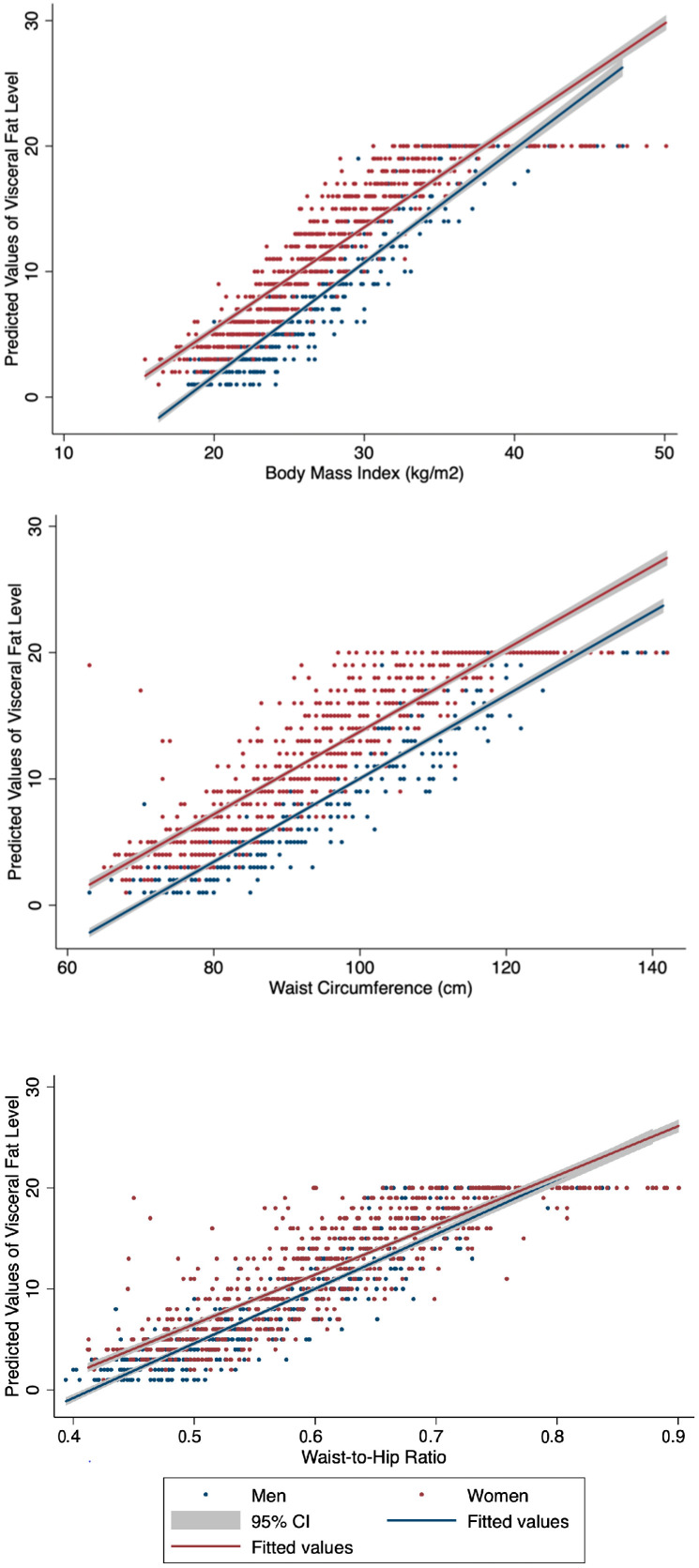
Predictive capacities of anthropometric indices of body fat distribution for the identification of visceral adiposity by sex among Nunavimmiut aged 16 and over, *Qanuilirpitaa*? 2017 Health Survey.

**Table 1 nutrients-16-00725-t001:** Population descriptive characteristics.

		Male		Female		*p*-Value
N		453		843		
Age	n (%)	n	%	n	%	
	16–19 years	88	30.0	136	26.6	0.2
	20–30 years	92	20.3	232	27.5	<0.05
	31–39 years	62	13.7	112	13.3	0.9
	40–44 years	34	7.5	81	9.5	0.2
	45–49 years	38	8.4	49	5.8	0.1
	50–54 years	35	7.7	83	9.7	0.3
	55–59 years	33	7.3	61	7.5	0.9
	60–65 years	34	7.5	38	4.5	<0.05
	65+ years	37	8.2	51	6.0	0.2
Weight	kg	73.8	17.9	67.6	16.9	<0.05
Height	cm	166.9	6.6	155.3	6.1	<0.05
BMI	kg/m^2^	26.31	5.7	28.0	6.4	<0.05
Waist circumference	cm	91.9	16.6	94.0	16.1	<0.05
Waist to Height ratio		0.55	0.10	0.61	0.10	<0.05
Body Fat Mass	kg	18.15	12.2	26.06	12.35	<0.05
Percentage of body fat	%	22.73	10.0	36.76	9.56	<0.05
Skeletal Muscle Mass		31.05	4.8	22.4	3.5	<0.05
Visceral fat level *		5	3–11	12	6–17	<0.05
High visceral fat level ^1^		91	20.09	459	54.44	<0.05
Basal metabolic rate	kcal/day	1566.43	170.03	1260.29	125.44	<0.05
Blood pressure						
Systolic	mmHg	128	13	120	14	<0.05
Diastolic	mmHg	80	10	75	9	<0.05
Smoking status	n (%) No	113	25.0	169	20.0	0.08
	Occasional	41	9.1	70	8.3	<0.05
	Daily	292	64.5	592	70.2	<0.05
Alcohol consumption	n (%)					
	Daily	19	4.2	39	4.6	<0.05
	3 to 6 per week	50	11.0	119	14.1	<0.05
	1–2 per week	96	21.1	169	20.0	<0.05
	1–3 per month	105	23.2	176	20.6	<0.05
	<1 a month	104	22.9	142	16.6	<0.05
	Never	55	12.1	140	16.4	<0.05
Degree of adiposity	n (%)					
	Overweight	110	24.3	203	24.1	<0.05
	Obesity	94	20.8	217	25.7	<0.05
	Severe Obesity	22	4.9	144	17.1	<0.05
Abdominal obesity	n (%)	210	46.4	664	78.8	<0.05
Dyslipidemia	n (%)	164	36.2	249	29.5	<0.05
Insulin resistance	n (%)	134	55.4	709	67.3	0.9
Pre-diabetes	n (%)	41	9.1	84	10.0	<0.05
Type 2 diabetes	n (%)	27	6.0	62	7.3	<0.05
Chronic renal disease (EGFR ≤ 90 mL/min)	n (%)	119	26.3	145	17.2	<0.05
Hypertension	n (%)	117	25.8	126	14.9	<0.05
Non-alcoholic Fatty Liver Disease ^2^	n (%)	14	3.1	20	2.4	0.6
Advanced stage of liver fibrosis ^3^	n (%)	1	0.2	1	0.1	0.7

Abbreviation: EGFR, Estimated glomerular filtration rate; BMI, body mass index. All normally distributed variables are presented as mean ± SD. Non-normally distributed variables (*) presented as median (IQR 25–75%). ^1^ Defined as a score ≥ 13. ^2^ Defined as a FLI score ≥ 60. ^3^ Defined as a FIB-4 score > 2.67.

**Table 2 nutrients-16-00725-t002:** Biomarker profile among Nunavimmiut 16 years and over, *Qanuilirpitaa*? 2017 Health Survey.

		Male		Female		*p*-Value
N		453		843		
White blood count	WBC per microliter	7.70	2.13	7.53	2.16	0.16
Hemoglobin	nmol/L	142.55	12.13	128.37	11.11	<0.05
Platelets	Platelets per microliter	305.28	66.82	346.85	69.26	<0.05
Creatinine	mg/dL	83.02	15.42	61.24	11.97	<0.05
24 H micro-albuminuria	mg/24 h	16.41	35.91	24.41	86.58	0.06
EGFR		101.38	19.97	107.59	19.18	<0.05
GGT	U/L	33.51	37.77	28.93	31.67	<0.05
ALT	U/L	12.71	10.19	11.66	12.27	0.1
AST	U/L	24.35	44.22	20.25	15.81	<0.05
ALB	g/L	44.35	3.00	43.64	2.96	<0.05
Bilirubin	μmol/L	5.60	3.25	4.17	2.16	<0.05
Apolipoprotein B	mg/dL	0.89	0.27	0.89	0.25	0.6
TG	mmol/L	1.62	1.78	1.47	0.98	0.06
LDL-Chol	mmol/L	2.58	0.93	2.64	0.83	0.2
TOTAL Chol	mmol/L	4.73	1.09	4.97	1.06	<0.05
HDL-Chol	mmol/L	1.43	0.42	1.63	0.50	<0.05
Total Chol/HDL Ratio		3.58	1.62	3.24	1.06	<0.05
HbA1C	%	5.56	0.60	5.47	0.74	<0.05
Random glucose	mmol/L	5.70	1.32	5.75	1.83	0.6
Fasting glucose	N	206		329		
	mmol/L	5.50	0.76	5.36	0.70	<0.05
Fasting insulin ^1^	N	213		336		
	mmol/L	48.25	30.37–79.71	60.53	41.56–92.98	0.2
HOMA-IR ^1^	N	206		329		
		2.83	2.94	3.06	2.77	0.4

All normally distributed variables are presented as mean ± SD. ^1^ Presented as median (IQR 25–75%). Abbreviations: WBC, white blood cell; EGFR, estimated glomerular filtration rate; HDL-Chol, high-density lipoprotein; LDL-Chol, low-density lipoprotein; TG, triglycerides; ALT, alanine aminotransferase; AST, aspartate aminotransferase; GGT, gamma-glutamyl transferase; ALB, albumin; HbA1c, glycated hemoglobin; HOMA-IR, homeostasis model assessment of insulin resistance.

**Table 3 nutrients-16-00725-t003:** Predictive capacities of anthropometric indices and VFL to detect cardiometabolic disorders among Nunavimmiut aged 16 and over by sex, *Qanuilirpitaa*? 2017 Health Survey.

	Weight			BMI			WC			WHtR			VFL		
	AUROC	95% IC		AUROC	95% IC		AUROC	95% IC		AUROC	95% IC		AUROC	95% IC	
**Male**															
Diabetes	0.75	0.63	0.86	0.78	0.70	0.87	0.77	0.66	0.88	0.79	0.67	0.89	0.77	0.66	0.87
Prediabetes	0.71	0.62	0.80	0.74	0.66	0.82	0.75	0.66	0.84	0.76	0.68	0.85	0.74	0.65	0.83
Insulin resistance	0.75	0.70	0.79	0.73	0.68	0.78	0.73	0.68	0.78	0.70	0.65	0.75	0.71	0.66	0.76
Hypertension	0.70	0.65	0.75	0.73	0.69	0.78	0.75	0.71	0.80	0.76	0.72	0.80	0.74	0.70	0.79
Dyslipidemia	0.71	0.66	0.76	0.73	0.68	0.79	0.75	0.70	0.79	0.75	0.70	0.79	0.75	0.70	0.80
**Female**															
Diabetes	0.66	0.57	0.76	0.71	0.62	0.81	0.66	0.55	0.78	0.68	0.56	0.80	0.69	0.60	0.79
Prediabetes	0.64	0.57	0.70	0.70	0.64	0.76	0.68	0.62	0.75	0.71	0.65	0.78	0.69	0.63	0.76
Insulin resistance	0.71	0.67	0.76	0.72	0.68	0.77	0.72	0.67	0.76	0.71	0.66	0.76	0.70	0.66	0.75
Hypertension	0.64	0.59	0.68	0.65	0.61	0.70	0.66	0.62	0.70	0.66	0.61	0.70	0.66	0.62	0.70
Dyslipidemia	0.65	0.61	0.70	0.69	0.65	0.74	0.70	0.67	0.75	0.72	0.68	0.76	0.69	0.64	0.73

Abbreviations: BMI, body mass index; WC, waist circumference; WHtR, waist-to-height ratio; VFL, visceral fat level; AUROC, Area-Under-the-Operating-Curve; IC, Interval Confidence

## Data Availability

The survey data are owned by Inuit and can be accessed through a request made to Qanuilirpitaa? 2017 DMC (nunavikhealthsurvey@ssss.gouv.qc.ca).

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
