# Peer review of "Adiposity Phenotypes and Associated Cardiometabolic Risk Profile in the Inuit Population of Nunavik"

_nutrients, 2024, doi:10.3390/nu16050725_

Round 1

Reviewer 1 Report

Comments and Suggestions for Authors

The authors conducted an observational study to examine the association of adiposity phenotypes with cardiometabolic risk factors in the Inuit population of Nunavik. By recruiting a sample of 1296 Inuit, the authors showed the prevalence of obesity and abdominal obesity in this sample. They showed that abdominal obesity, defined based on waist circumference, waist-height ratio, or visceral fat level, was variably associated with cardiometabolic risk factors. These findings are interesting with potential implications. There are some comments.

1.     Methodology (line 112 on page 3): “Potential participants were selected from a proportional sampling of the Makivvik Corporation’s beneficiary list -.” Please provide more details on the Makivvik Corporation’s beneficiary list. Notably, it is unclear whether all (or almost all) Inuits were on the list. If not, there might be selection bias issues if the potential participants were entirely selected from the list, and a discussion of this issue is recommended in the Discussion.

2.     Methodology: The author described the exclusion criteria (line 116). Please define the inclusion criteria.

3.     Methodology (line 115 on page 3): “A total of 1326 individuals participated in the study-.” However, it seems that there were only 1296 participants. Please clarify.

4.     Methodology: If the information is available, I recommended that the authors present the number of potentially eligible participants, the number of potentially eligible participants assessed for eligibility, the number of participants who were confirmed eligible, and the number of participants who were confirmed eligible and agreed to participate.

5.     Methodology (statistical analysis): The authors applied linear regression analysis. Please describe whether the assumptions (for instance, the normality assumption and the linearity assumption) were tested and whether the assumptions were met.

6.     Results (Figure 2): The authors presented the predicted VFL of different BMI and waist circumference levels. Please also show the confidence intervals of the predicted VFL.

7.     Discussion: The authors observed that the adiposity phenotypes, defined based on waist circumference, waist-height ratio, or visceral fat level, might be variably associated with a cardiometabolic risk factor (for instance, diastolic blood pressure in Supplementary Table 2). Also, an adiposity phenotype (for instance, waist circumference) might be variably associated with cardiometabolic risk factors (for instance, waist circumference might be associated with triglyceride but not cholesterol levels). A discussion of these findings is recommended.

8.     Discussion: Although there were recommended cut-offs for defining adiposity phenotypes (for instance, weight circumference), the authors decided to analyze the adiposity phenotypes as continuous variables (for instance, Supplementary Table 2). A discussion of this decision is recommended. For instance, the rationale might be that the optimal cut-off in the Inuit population remains unclear.  

Author Response

Dear editorial committee

We would like to express our sincere gratitude to the reviewers for their thorough and insightful evaluation of this manuscript. Their valuable feedback has significantly enriched the quality of the paper, and we are truly appreciative of the time and expertise they have dedicated to this review.

We have made further changes to the original manuscript in response to the reviewers’ comments and have shown all additional text using track changes in the revised manuscript. We provide detailed responses to each of the comments below and indicate where the changes to the manuscript can be found. We believe that we have answered all the points raised, and hope that you now find our manuscript acceptable for publication.

Please find attached to this note specific answers to your comments, 

Thank you and best regards,

Dr Pierre Ayotte

Reviewer 2 Report

Comments and Suggestions for Authors

The authors try to explore the impact of the prevalence of obesity on their cardiometabolic health in the Inuit Population of Nunavik. However, there  are several issues need to be addressed. 

Major issues:

1. The age range is too large, and age factors are related to the incidence of cardiovascular diseases. In addition, the number of subjects in each age group was small. It is suggested to compare the incidence and gender differences of cardiovascular metabolic diseases in different age groups.

2. Why was not the hyperuricemia included in the cardiometabolic risk factors?

3. It is recommended to specify the measurement standards of blood pressure, fasting time of fasting blood glucose, and the measurement and scoring standards of visceral fat levels.

4. It is suggested that the authors provide detailed inclusion and exclusion criteria.

Minor issues:

1. Graphic aesthetics need to be improved.

2. The Supplementary Table 6 and Supplementary Table 7 were missed in the Supplements.

Author Response

Dear editorial committee

We would like to express our sincere gratitude to the reviewers for their thorough and insightful evaluation of this manuscript. Their valuable feedback has significantly enriched the quality of the paper, and we are truly appreciative of the time and expertise they have dedicated to this review.

We have made further changes to the original manuscript in response to the reviewers’ comments and have shown all additional text using track changes in the revised manuscript. We provide detailed responses to each of the comments below and indicate where the changes to the manuscript can be found. We believe that we have answered all the points raised, and hope that you now find our manuscript acceptable for publication.

Please find attached specific answers to your comments, 

Thank you and best regards,

Dr Pierre Ayotte

Reviewer 3 Report

Comments and Suggestions for Authors

Summary:

This study utilised data collected in 2017 Nunavik Inuit Health survey to investigate body composition and the relationship between anthropometric and cardiometabolic parameters among an Inuit population, who are underrepresented in the literature.

The study has strengths in terms of the focus on this underrepresented population and provides a strong array of cardiometabolic measures. There are also inherent limitations with various instruments and classifications used for body composition that have not been previously validated in this population. Also the data was collected seven years ago and the delay in reporting these findings is unclear.

Prior to being considered for publication, there are various comments to address.

Comments:

Abstract

-          Check first authors name for error

-          Line 24: Check the use of “respectively”, it seems incorrect

-          Line 31: “inconstantly” should be amended to “inconsistently” if appropriate?

Introduction

-          Line 37: The citations provided relate to obesity prevalence data that is 20+ years old, are there any other more recent data that you can cite?

-          Line 39-42: This sentence is long, consider breaking it up

-           

Materials and Methods

-          Line 124: How was the device validated?

-          Line 128: VFL has been abbreviated earlier

-          Line 132: Why is there a “t” in WHR abbreviation?

-          Line 152: Grammar check

-           

Data Analyses

-          Line 170: How were data assessed to check they met assumptions for parametric data analysis?

Results

-          Line 191: Grammar check

-          Line 217: This detail should just be described in your statistical analysis section

-          Line 237: Ensure all specific correlation values and corresponding significance values are presented in text, where not featuring in your tables

-          Line 260: This should be “population descriptive characteristics” since this is a cross-sectional study.

-          Line 260: It would be useful to add skeletal muscle mass descriptive data to the table

Discussion

-          Line 422: It should also be noted that that the use of other cut points, such as that for waist circumference were not validated for Inuit

-          Line 423: It would be logical here or in discussion to provide recommendations for future research to conduct validation studies for cardiometabolic parameters among this population as it has been neglected from existing research

-          Line 359: This statistic differs from what you reported on line 204. Why is this?

Comments on the Quality of English Language

Please check grammar and spelling throughout as there are various errors present

Author Response

Dear editorial committee

We would like to express our sincere gratitude to the reviewers for their thorough and insightful evaluation of this manuscript. Their valuable feedback has significantly enriched the quality of the paper, and we are truly appreciative of the time and expertise they have dedicated to this review.

We have made further changes to the original manuscript in response to the reviewers’ comments and have shown all additional text using track changes in the revised manuscript. We provide detailed responses to each of the comments below and indicate where the changes to the manuscript can be found. We believe that we have answered all the points raised, and hope that you now find our manuscript acceptable for publication.

Please find attached our specific answers to your comments, 

Thank you and best regards,

Dr Pierre Ayotte
